# Homo- and heterodehydrocoupling of phosphines mediated by alkali metal catalysts

Lipeng Wu [1,2,6], Vincent T. Annibale [1,6], Haijun Jiao[3], Adam Brookfield[4], David Collison[4] & Ian Manners[1,5]

Catalytic chemistry that involves the activation and transformation of main group substrates is relatively undeveloped and current examples are generally mediated by expensive transition metal species. Herein, we describe the use of inexpensive and readily available *t*BuOK as a catalyst for P–P and P–E (E = O, S, or N) bond formation. Catalytic quantities of *t*BuOK in the presence of imine, azobenzene hydrogen acceptors, or a stoichiometric amount of *t*BuOK with hydrazobenzene, allow efficient homodehydrocoupling of phosphines under mild conditions (e.g. 25 °C and < 5 min). Further studies demonstrate that the hydrogen acceptors play an intimate mechanistic role. We also show that our *t*BuOK catalysed methodology is general for the heterodehydrocoupling of phosphines with alcohols, thiols and amines to generate a range of potentially useful products containing P–O, P–S, or P–N bonds.

[1] School of Chemistry, University of Bristol, Cantock's Close, Bristol BS8 1TS, UK. [2] State Key Laboratory for Oxo Synthesis and Selective Oxidation, Suzhou Research Institute of Lanzhou Institute of Chemical Physics, CAS, 730000 Lanzhou, P. R. China. [3] Leibniz-Institut für Katalyse e. V, Albert-Einstein-Straße 29a, 18059 Rostock, Germany. [4] The School of Chemistry and the Photon Science Institute, The University of Manchester, Oxford Road, Manchester M13 9PL, UK. [5] Department of Chemistry, University of Victoria, Victoria, BC V8W 3V6, Canada. [6] These authors contributed equally: Lipeng Wu, Vincent T. Annibale. Correspondence and requests for materials should be addressed to I.M. (email: imanners@uvic.ca)

W hen compared to the traditional stoichiometric salt metathesis and reductive coupling reactions that still dominate the formation of element–element bonds in main group chemistry, catalytic methods represent a highly attractive alternative synthetic approach[1–7]. For example, substantial advances have been made in catalytic dehydrocoupling of *p*-block substrates to form both homonuclear (E–E) or heteronuclear (E–E′) bonds (E, E′ = *p*-block element). However, most of the current catalysts in use are based on precious transition metals such as Rh[8–11], Ir[12–15] and Ru[16] raising concerns about their high price as a result of their low natural abundance and their presence as potentially toxic residues in polymer products[17–19]. Although significant progress has been made in terms of the use of earth abundant metals such as Zr[20–22], Fe[23,24] and Ni[25,26], the development of transition metal-free dehydrocoupling catalysts also offers considerable potential. For example, recent reports describe advances concerning the use of main group species such as [(Dipp-nacnac)Mg$^n$Bu]$_2$[27], Al(NMe$_2$)$_3$[28] and B(C$_6$F$_5$)$_3$[29], especially in the areas of N–B and P–Si bond formation. Rare examples of the use of alkali metal reagents such as KN(SiMe$_3$)$_2$[30], 1-lithium-2-*tert*-butyl-1,2-dihydropyridine[31], or group 1 salts containing a carbazolido NNN pincer ligand[32] for the dehydrocoupling of Me$_2$NH•BH$_3$ to form [Me$_2$NBH$_2$]$_2$ have also been described.

Molecular compounds containing P–P bonds have numerous applications in coordination[7,33,34] and synthetic chemistry[29,35–37], and diphosphines readily react with alkenes or alkynes to form bidentate ligands[38–40]. Probably because of the potential catalyst deactivation due to unproductive phosphine coordination, only few transition metal catalysts (Zr, Rh) have been successfully applied for P–P bond formation via catalytic dehydrocoupling[3,7]. Furthermore, in most cases, relatively high temperatures (110–140 °C) and long reaction times (3–4 days) were required[2,41–44]. Recently, transition metal-free catalysts (C$_5$Me$_5$)$_2$SnCl$_2$[45], B(*p*-C$_6$F$_4$H)$_3$[46] and stoichiometric reagents, namely lithium chloride carbenoids[47] and *N*-heterocyclic carbenes[48], have also been reported to mediate phosphine dehydrocoupling although either harsh reaction conditions, long reaction times, or non-commercially available reagents were still required (Fig. 1a).

In addition to species containing P–P bonds, compounds containing P–O, P–S and P–N bonds also have widespread importance in catalysis, organic synthesis, and in biochemistry and agrochemistry[49,50]. Apart from an example of heterodehydrocoupling of phosphines with protic substrates (PhS-H) to form P–S bonds mediated by in situ generated rhodium phosphido complexes at 110 °C (Fig. 1b)[2], catalytic heterodehydrocoupling of phosphines with other protic substrates is unexplored. In particular, the development of a cost-effective, general, and practical transition metal-free catalytic homodehydrocoupling and heterodehydrocoupling of phosphines is highly desirable. Herein, we report our findings on the use of inexpensive and commercially available reagent *t*BuOK as a general catalyst for dehydrocoupling reactions involving phosphines and the formation of P–P, P–O, P–N and P–S bonds (Fig. 1c).

## Results

**Dehydrocoupling of phosphines**. Phosphorus and hydrogen possess similar Pauling electronegativities leading to only a weak polarisation for P–H bonds. As strong Lewis acids such as B(*p*-C$_6$F$_4$H)$_3$ have been shown to dehydrocouple phosphines (R$^1$R$^2$PH, R$^2$ = R$^1$ or H) via a mechanism that involves sequential hydride and proton removal from phosphorus[46], we rationalised that a strong base might be able to mediate the same process.

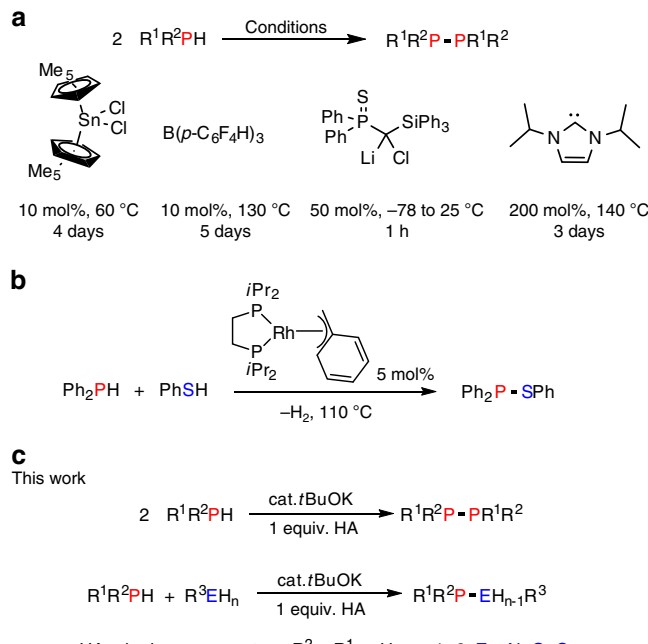

**Fig. 1** Developed catalysts for P–P and P–E bond formation. **a** Main group catalysts and conditions for the homodehydrocoupling of phosphines. **b** Heterodehydrocoupling of diphenylphosphine with thiophenol by in situ generated rhodium phosphido complexes. **c** Our current work using *t*BuOK as the catalyst for the homo- and heterodehydrocoupling of phosphines

Based on the recently revealed, highly versatile role of *t*BuOK in initiating C–C[51,52], C–Si,[53] Si–O[54] and C–N bond formation,[55] we studied the use of *t*BuOK as a catalyst for the homodehydrocoupling of phosphines. Initially, we chose diphenylphosphine (**1a**) as a model substrate and the reaction was attempted with 0.1 mmol of **1a** with 10 mol% *t*BuOK in 0.5 mL THF at 130 °C. After 16 h no reaction had taken place. Given the recent examples of the introduction of a hydrogen acceptor (**HA**) for the promotion of dehydrocoupling reactions[29,46,56,57], we then explored the use of different types of **HA** for this reaction with dramatically different results (Fig. 2a and Supplementary Table 1). **HA-1** to **HA-5** were most effective: for example, benzophenone (**HA-1**) led to 86% conversion of **1a** and the formation of **2a** in 47% yield whereas *trans*-stilbene (**HA-3**) gave full conversion of **1a** accompanied by the production of **2a** in 69% yield. In the latter case residual **1a** was found to be converted to the hydrophosphination product. Most impressively, with the addition of *N*-benzylideneaniline (**HA-5**) we achieved >99% conversion of **1a** with the formation of **2a** in 92% yield. Computationally, the parent dehydrocoupling reaction of 2 equiv. **1a** to **2a** and H$_2$ in the absence of an **HA** was thermodynamically uphill by 2.64 kcal mol$^{-1}$, and likely possesses a significant kinetic barrier. The dehydrocoupling reactions with added **HA-1** to **HA-5** were all calculated to be thermodynamically exergonic (see Supplementary Discussion and Supplementary Data 1, 2 in the Supplementary Information).

With **HA-5**, efforts to perform the reaction under milder conditions were made using different bases, temperatures, and catalyst loadings (Supplementary Tables 2–5). Notably, with 0.01 mmol of *t*BuOK (10 mol%) and 0.1 mmol of **HA-5**, 82% yield of **2a** was obtained after 64 h at 25 °C (Supplementary Table 2, entry 4). Even with 2.5 mol% *t*BuOK at 60 °C, we could still achieve 85% yield after 24 h (Supplementary Table 2, entry 6). It is worth noting that 10 mol% of different alkali metal *tert*-butoxides were

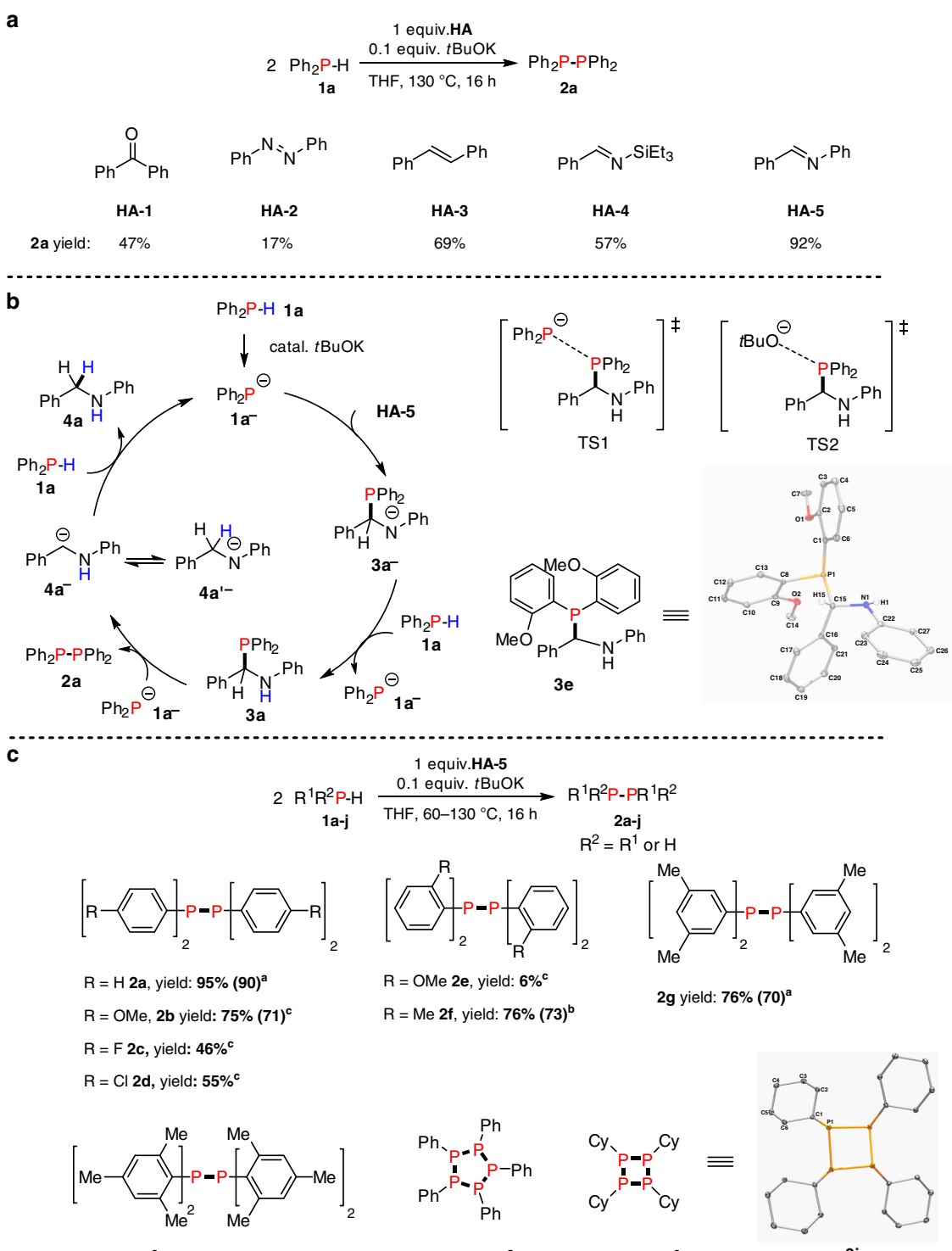

**Fig. 2** *t*BuOK-catalysed dehydrocoupling of phosphines in the presence of HA. **a** The effect of selected **HA** in the homodehydrocoupling of Ph₂PH. **b** Proposed reaction mechanism via **3a** as key intermediate and S_N2-type reaction to produce **2a**, K⁺ ions are removed from the catalytic cycles for clarity. Proposed transition state structures TS1, and TS2, for attack of **3a** by incoming **1a⁻** or *t*BuO⁻. X-ray crystal structure of hydrophosphination product **3e** with non-hydrogen atoms shown as 30% probability ellipsoids. **c**, Reaction generality using different phosphines: reactions were performed with 0.1 mmol phosphines, 0.01 mmol *t*BuOK, 0.1 mmol **HA-5**, 0.5 mL THF in a J. Young NMR tube and yields were based on the phosphine as the limiting reagent and determined by ³¹P{¹H} NMR spectroscopy using a capillary of PCl₃ as a calibration standard, the numbers in brackets were isolated yields of products. [a]60 °C, [b]100 °C, [c]130 °C. X-ray crystal structure of **2j** with non-hydrogen atoms shown as 30% probability ellipsoids

screened as catalysts for the dehydrocoupling reaction with added **HA-5** where the activity followed the order of K>Na>Li (Supplementary Table 4). We then chose 60 °C and 10 mol% *t*BuOK as the reaction conditions (which gave a 95% yield of **2a** after 8 h, Supplementary Table 4) for further mechanistic studies. It is known that P–H bonds can be activated by strong bases to initiate the subsequent hydrophosphination of the C=N or C=C double bond[58–60]. We explored the potential role of the hydrophosphination product from **1a** and **HA-5** and carried out several control experiments; the results are summarised in Supplementary Fig. 1. Heating **1a** or a mixture of **1a** and **HA-5** at 60 °C in THF gave no reaction after 8 h (Supplementary Fig. 1, Eqs. 1–2). When 10 mol% of *t*BuOK was added to the THF solution containing **1a** and **HA-5**, the rapid emergence of a new peak at 3.3 ppm in the $^{31}$P{$^1$H} nuclear magnetic resonance (NMR) spectrum was observed (Supplementary Fig. 2). This was assigned to the hydrophosphination adduct **3a** upon synthesis and subsequent characterisation of a sample of this species (Supplementary Fig. 1, Eq. 3). After 8 h compound **3a** was quantitatively converted to a mixture of the homodehydrocoupling product **2a**, and *N*-benzylaniline **4a**, and **HA-5** in a 1:1:1 molar ratio (Supplementary Fig. 1, Eq. 5 and Supplementary Figs. 3–5). Monitoring the transformation of **3a** to **2a** at 30 min intervals showed that no other *P*-containing species were involved, except for minor amounts of **5a** by-product which were also observed (Supplementary Fig. 6). It is worth noting that **3a** is in equilibrium with **HA-5** and **1a** (Supplementary Fig. 1, Eq. 4), and that the transformation of **3a** to the 1:1:1 mixture of **2a**, **4a**, and **HA-5** requires the presence of the *t*BuOK catalyst (Supplementary Fig. 1, Eq. 5)[61]. Our experiments suggest that **3a** is the key intermediate for the homodehydrocoupling of **1a**.

Based on the results of the stepwise experiments we proposed the reaction mechanism outlined in Fig. 2b. Firstly, **1a** is deprotonated by *t*BuOK to form diphenylphosphide **1a**$^−$. This activation facilitates the addition of **1a** to **HA-5** to form the hydrophosphination adduct **3a**. Based on the knowledge that S$_N$2 reactions at phosphorus are possible[62–64] and similar halophilic reactions exist, we anticipated that hydrophosphination adduct **3a** is attacked by anionic **1a**$^−$ at the phosphorus centre through a S$_N$2-type of reaction (TS1) which eventually yields **2a**. Similarly, **3a** could be attacked by the *t*BuO$^−$ anion (TS2) which leads to the side product Ph$_2$P-O*t*Bu **5a** detected by $^{31}$P{$^1$H} NMR as a peak at 86.9 ppm (Supplementary Fig. 6). Here we propose a carbanionic leaving group (**4a**$^−$) which may tautomerise to the amide anion (**4a**$'^−$), which can also deprotonate **1a** and regenerate **1a**$^−$ and close the catalytic cycle. Similar transient carbanionic species are also proposed in Brook rearrangements of silylated amines in the presence of a catalytic amount of base[65,66], and carbanionic leaving groups are also known in reactions of phosphine oxides with organometallic reagents[67]. Another possibility is following the nucleophilic attack of the phosphorus of **3a** by an anion such as **1a**$^−$ or *t*BuO$^−$ there may be a process in which the P–C bond cleavage process is accompanied by protonation by an incoming protic substrate at the incipiently generated and partially carbanionic site.

When **HA-5** was substituted at the *para*-position of the benzylidene phenyl group with an electron-withdrawing group (–COOMe) the initial reaction rate was substantially increased, on the other hand with an electron-donating (–Me) group the initial reaction rate was much slower (Supplementary Fig. 7). These kinetic observations provide further support for the proposed reaction mechanism described in Fig. 2b as an electron-withdrawing group should stabilise the anionic leaving group **4a**$^−$.

The substrate generality for the reaction was then studied (Fig. 2c). With diphenyl phosphine (**1a**) as substrate, a 95% yield

of the dehydrocoupling product **2a** could be obtained. In the case of *para*-methoxy-substituted secondary phosphines, a 75% yield of **2b** was gained. For chloro- and fluoro-substituted phosphines, more moderate yields were observed for **2c** and **2d**, perhaps due to the side reactions involving C–F and C–Cl bonds such as dehalogenation[68,69]. When bis(*ortho*-methoxyphenyl) phosphine was used we can only obtain 6% of the homodehydrocoupling product **2e** due to the existence of a large quantity of intermediate **3e**, which was crystallographically characterised (Fig. 2b, Supplementary Table 7), which suggests that the sterics of the substituents at phosphorus play a key role and is consistent with our proposed mechanism involving both *P*-based nucleophiles and electrophiles. Thus, for example, with less bulky *meta*-methylphenyl substituents on phosphorus a 76% yield of **2g** was observed. On the other hand, the presence of a bulkier mesityl substituent (as in **2h**) totally suppressed the hydrophosphination and the subsequent dehydrocoupling reaction. We also extended our protocol to primary phosphines and found that phenylphosphine was selectively transformed into 5-membered cyclic ring [PhP]$_5$ **2i** in 63% yield. Interestingly, when the aliphatic primary phosphine cyclohexylphosphine was used, cyclic **2j**, which contains a 4-membered ring, was produced in 29% yield and this species was characterised crystallographically (Fig. 2c, Supplementary Table 7).

Based on our proposed mechanism in Fig. 2b, the high conversion of **1a** but lower yield of diphosphine **2a** for the cases of **HA-3** and **HA-4** can be explained by the presence of unconverted hydrophosphination intermediate (Supplementary Figs. 10, 11). However, in the case of azobenzene (**HA-2**), where the yield of **2a** is particularly low (17%), no hydrophosphination product was detected at 130 °C in THF over 16 h. Instead, in addition to the peak for **2a**, two further peaks were present in the $^{31}$P{$^1$H} NMR spectrum and these were assigned to Ph$_2$P-O*t*Bu (**5a**, 86.9 ppm) and Ph$_2$P-NHPh (**6a**, 26.9 ppm) (Supplementary Fig. 9). We then explored whether the reaction using **HA-2** could be further optimised (Supplementary Table 6). Remarkably, on lowering the reaction temperature from 130 to 25 °C while using 10 mol% of *t*BuOK resulted in the formation of **2a** in 75% yield within 5 min (Supplementary Table 6, entry 1).

Low-temperature $^{31}$P NMR spectroscopy was performed in THF to provide some initial insight into the reaction (Supplementary Fig. 13). Gradually increasing the temperature from −60 °C to room temperature clearly showed the decay of **1a** and growth of the diphosphine **2a**. At 25 °C, two additional peaks at 60.8 ppm (hydrophosphination adduct Ph$_2$P-N(Ph)-NHPh, **7a**)[70] and 86.9 ppm (Ph$_2$P-O*t*Bu, **5a**) appeared. Further experiments showed that when the stoichiometry of the reaction was changed and a mixture composed of 2 equiv. of **1a** and 1 equiv. of **HA-2** was treated using 10 mol% of *t*BuOK resulted in complete conversion to a 1:1 mixture of **2a** and hydrazobenzene within 5 min at 25 °C (Fig. 3a, Supplementary Figs. 14 and 15).

A series of stoichiometric and catalytic reactions were performed in an attempt to determine potential active species present over the course of catalysis. In the 1:1 stoichiometric reaction of orange azobenzene (**HA-2**) with *t*BuOK in THF-*d*$_8$ a dark brown reaction mixture resulted for which analysis by $^1$H and $^{13}$C{$^1$H} NMR revealed the presence of only unreacted diamagnetic **HA-2** and *t*BuOK (Supplementary Fig. 16). However a solution-state EPR spectrum in THF/2-Me-THF (20:1) at 25 °C revealed a signal at approximately *g* = 2 which is characteristic of an organic radical and shows a hyperfine structure consistent with the previously reported azobenzenyl radical anion K [PhNNPh] (Fig. 3b, Supplementary Fig. 16)[71,72]. The 1:1 stoichiometric reaction of **HA-2** with K[PPh$_2$] (**1a**$^−$) also yielded a deep

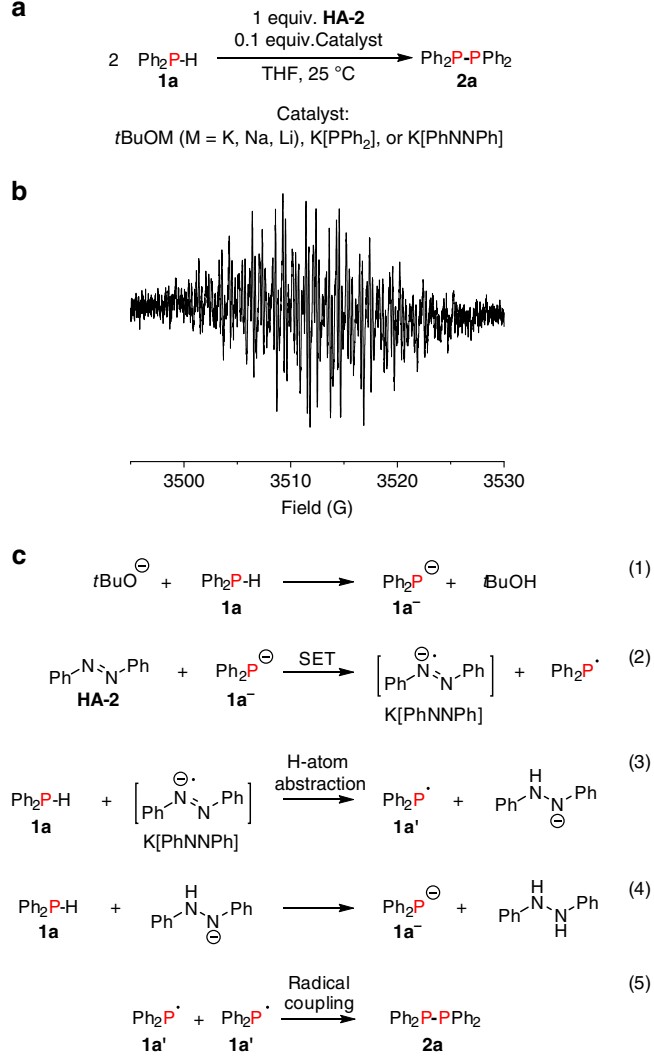

**Fig. 3** Dehydrocoupling of **1a** in presence of azobenzene (**HA-2**). **a** Homodehydrocoupling of **1a** catalysed by $t$BuOM (M = K, Na, Li), K[PPh$_2$] (**1a⁻**), or K[PhNNPh] at 25 °C. **b** X-band EPR spectrum of K[PhNNPh] in THF/2-Me-THF (20:1) generated in the 1:1 stoichiometric reaction of **HA-2** with $t$BuOK. **c** Proposed radical chain mechanism for dehydrocoupling of **1a** to **2a** mediated by in situ generated radical anion K[PhNNPh]

brown solution however by $^1$H and $^{31}$P{$^1$H} NMR the only diamagnetic species observed was **2a**, as **HA-2** was fully converted to its radical anion K[PhNNPh], which was also observed by EPR (Supplementary Fig. 17).

The radical anion K[PhNNPh] could also be independently prepared from potassium metal and **HA-2** for use in further studies on its role as a catalyst (Supplementary Fig. 18). Upon the addition of 10 mol% K[PhNNPh] to a 1:1 stoichiometric mixture of **1a** and **HA-2** gave approximately a 1:1 mixture of **7a** and **2a** by $^{31}$P{$^1$H} NMR spectroscopy along with hydrazobenzene (Supplementary Fig. 19). Remarkably, treatment of a mixture of 2 equiv. of **1a** and 1 equiv. of **HA-2** in THF with either 10 mol% **1a⁻** or independently synthesised K[PhNNPh] resulted in complete conversion to a 1:1 mixture of **2a** and hydrazobenzene within 5 min at 25 °C (Fig. 3a, Supplementary Figs. 14 and 15), analogous to the results obtained when 10 mol% $t$BuOK was used. The K[PhNNPh] radical anion was also observed by EPR spectroscopy in the reaction of **HA-2** with $t$BuOK and 0.5 equiv. of the substrate **1a** (Supplementary Fig. 20).

In addition, the use of 10 mol% $t$BuONa also resulted in complete conversion of 2 equiv. of **1a** and 1 equiv. of **HA-2** within 5 min at 25 °C, while $t$BuOLi was significantly slower requiring 6 h at 25 °C to reach full conversion, likely due to the poor solubility of $t$BuOLi in THF (Fig. 3a). While both **1a⁻** and K[PhNNPh] can catalyse the dehydrocoupling of **1a** to **2a**, inexpensive and commercially available $t$BuOK provides a convenient and practical entry point into the catalysis.

We also explored the reactivity of the hydrophosphination product **7a** which was observed as a $P$-containing product along with **2a** in reactions involving 10 mol% of an alkali metal-based catalyst ($t$BuOK, **1a⁻**, or K[PhNNPh]) and a 1:1 mixture of **1a** and **HA-2**. Treatment of **7a** with 1 equiv. of $t$BuOK or **1a⁻** also resulted in conversion to **2a**, hydrazobenzene, and K[PhNNPh] (Supplementary Fig. 21). In addition, treatment of a pure sample of independently synthesised **7a** with 10 mol% $t$BuOK resulted in ~43% conversion to **2a** and hydrazobenzene and ~7% conversion to **5a** within 5 min at 25 °C (Supplementary Fig. 22). Similar product distributions were also obtained using either 10 mol% **1a⁻** or K[PhNNPh] as catalysts (Supplementary Figs. 23 and 24). Interestingly the reaction of a 1:1 stoichiometric mixture of **1a** and **7a** with 10 mol% $t$BuOK or K[PhNNPh] resulted in complete conversion to **2a** and hydrazobenzene (Supplementary Figs. 25 and 26).

Radical trapping experiments with 1,4-cyclohexadiene, and experiments with added radical initiator di(t-butyl)peroxide (DTBP) were performed which are further supportive of a radical process (Supplementary Figs. 27 and 28). On the basis of the stoichiometric reactions and our key observation of the radical anion K[PhNNPh] and the demonstrated competency of this species in catalysis we propose a radical mechanism initiated by one-electron reduction of **HA-2** to produce the radical anion K[PhNNPh] (Fig. 3c).

The oxidation potential for $t$BuOK is at +0.10 V vs. SCE in DMF,[51] meanwhile the reduction potential for azobenzene (**HA-2**) is at −1.36 V vs. SCE in DMF.[73] Ashby and coworkers attempted to experimentally measure the oxidation potential of K[PPh$_2$] (**1a⁻**) in THF and were unsuccessful due to adsorption onto the Pt electrode, but concluded based on their studies on single electron transfer (SET) to organic iodides that they estimate the oxidation potential of **1a⁻** to be in the range of +0.8 to +1.3 V vs. SCE.[74] It seems unlikely that $t$BuOK acts directly as the primary electron donor to azobenzene given that there is a significant mismatch of redox potentials and in our experiments only a small amount of radical was produced in the 1:1 reaction of **HA-2** with $t$BuOK. It is more likely that **1a⁻** generated under the catalytic conditions from the deprotonation reaction of **1a** by $t$BuOK is engaged in electron transfer to **HA-2** to initiate the radical chain process (Fig. 3c). In addition, the diphenylphosphide anion has been shown to participate in SET chemistry to a range of organic molecules either under photoirradiation,[75,76] and without photoirradiation.[74,77,78]

It appears that in all reactions where an EPR signal was detectable that the only persistent radical species observed is the azobenzenyl radical anion K[PhNNPh], this assignment is on the basis of the $g$-value and coupling constants, which matched closely with the literature values[71,72] with varying degrees of spectral line broadening likely due to concentration and other solvent effects (see Supplementary Figs. 16–21 and 29). It is noteworthy that the $P$-based radical •PPh$_2$ (**1a′**) does not persist under our reaction conditions since previously the EPR spectrum for this unstable intermediate was collected at 77 K,[79] and therefore rapidly couples to give **2a** (Fig. 3c).

It is noteworthy that the hydrophosphination adduct **7a** can be converted both with and without a second equivalent of **1a** upon the addition of either catalytic or stoichiometric quantities of

*t*BuOK or **1a**⁻. This gave a mixture of **2a**, hydrazobenzene, and K [PhNNPh]. However, given that in the variable temperature NMR experiment (Supplementary Fig. 13) **7a** was not observed until the sample was warmed to 25 °C from −60 °C, it is likely that **7a** is an intermediate on a secondary pathway *en route* to **2a** and not an essential or central intermediate on the primary pathway to **2a**. This is in contrast to what was previously demonstrated using **HA-5** where the resulting hydrophosphination adduct **3a** was demonstrated to be a crucial intermediate.

A comparison between the use of *N*-benzylideneaniline (**HA-5**) and azobenzene (**HA-2**) was performed using different substituted phosphines (Fig. 4). For diphenylphosphine, although the yield of **2a** with **HA-2** was lower than for the case of **HA-5** (75% vs 95%), it was achieved at milder temperature (25 °C *vs* 60 °C) and with considerably shorter reaction time (<5 min vs. 8 h). From this point of view, **HA-2** is still a practical hydrogen acceptor. Thus, additional phosphine substrates were explored: we found that in some cases **HA-2** gave better conversion and yield of P–P coupled products than **HA-5** (Fig. 4). For example, with fluoro- and chloro-substituted phosphines, more than 30% yield increases (78% vs. 46% and 87% vs. 55%) were observed in each case at much lower temperature (−20 °C *vs* 130 °C) and short reaction time (<5 min vs. 120 h). For bis(mesityl)phosphine a 59% yield of diphosphine **2 h** was obtained using **HA-2**, while in the case of **HA-5**, no product was observed at all. The yield of [PhP]₅ (**2i**) was very high (92%) for the dehydrocoupling of phenylphosphine. When cyclohexylphosphine was used as a substrate, [CyP]₄ (**2j**) was produced in 66% yield.

**Heterodehydrocoupling of phosphines**. Next, given the virtual absence of previous examples[2], we investigated whether our *t*BuOK-catalysed dehydrocoupling protocol could be extended to the heterodehydrocoupling of phosphines with different protic substrates (R³E-H$_n$; E = N, O, S, n = 1, 2). The initial reaction was performed using 0.1 mmol Ph₂PH and 0.1 mmol *t*BuOH as substrates together with 0.1 mmol **HA-5** and 10 mol% *t*BuOK in 0.5 mL THF at 130 °C. We followed the reaction profile by ³¹P NMR spectroscopy at different time intervals (Fig. 5a); the imine hydrophosphination adduct **3a** is formed quantitatively after combining all the reagents. After heating the reaction mixture for 1 h we could observe both homodehydrocoupling product **2a** and heterodehydrocoupling product **5a**. Both **2a** and **5a** were present during the reaction and eventually all **2a** was converted to give **5a** in 90% yield together with some other minor side products. Computationally the heterodehydrocoupling reaction of **1a** with *t*BuOH in the presence of **HA-5** to produce **5a** and **4a** was exergonic by 10.62 kcal mol⁻¹, similar to the computed value of $\Delta G = -10.09$ kcal mol⁻¹ for the analogous homodehydrocoupling process (see Supplementary Discussion in the Supplementary Information and Supplementary Data 1 and 2). The similarly favourable thermodynamics calculated for the P–P and P–O coupling reactions suggest that both reactions are possible, and likely competitive. The further conversion of **2a** to **5a** in the presence of additional *t*BuOH allows for high yield of **5a**.

We then performed the heterodehydrocoupling reactions of secondary and primary phosphines with alcohols, amines or thiols in the presence of either azobenzene (**HA-2**) or imine (**HA-5**). As shown in Fig. 5b, we observed that **HA-2** was able to mediate the heterodehydrocoupling of diphenylphosphine with either *t*BuOH or aniline at 25 °C with very high yields (93% and 87%, respectively). At 25 °C with *p*-thiocresol there was no reaction; however, after heating at 130 °C for only 3 h, the heterodehydrocoupling product **8a** was obtained in 90% yield. Using **HA-5** on the other hand, all reactions involving diphenylphosphine required heating at 130 °C to obtain excellent

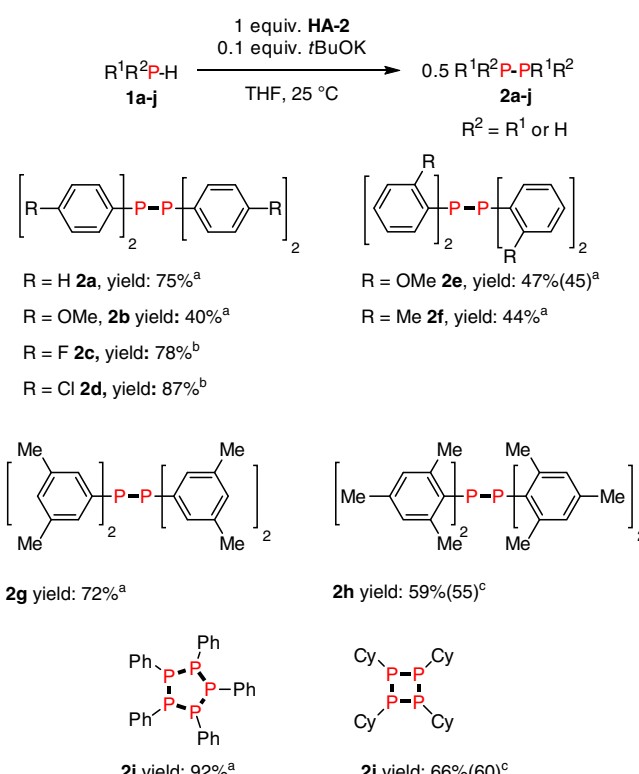

**Fig. 4** Reaction generality for the dehydrocoupling of different phosphines with HA-2: reactions were performed with 0.1 mmol phosphines, 0.01 mmol *t*BuOK, 0.1 mmol **HA-2**, 0.5 mL THF in a J. Young NMR tube and yields were based on the phosphine as the limiting reagent and determined by ³¹P {¹H} NMR spectroscopy using a capillary of PCl₃ as a calibration standard, the numbers in brackets were isolated yields. ᵃ25 °C within 5 min, ᵇ−20 °C within 5 min, ᶜ130 °C for 1 h

yields (90% and 85%) except for the case of reaction with aniline where only a moderate yield of **6a** (54%) was observed. When phenylphosphine was used as the substrate with **HA-2** the heterodehydrocoupling with *t*BuOH produced only 8% of the doubly-dehydrocoupled product **9a′**, while with **HA-5** the yields were 48% for singly-dehydrocoupled product **9a** and 20% for **9a′**. When phenylphosphine was reacted with aniline in the presence of **HA-2** we also observed the formation of only doubly dehydrocoupled **10a′** in 47% yield and with **HA-5** a 34% yield of **10a′** was obtained. The reaction with *p*-thiocresol mediated by **HA-2** produced 87% doubly-dehydrocoupled product **11a′** alongside only 8% of **11a**, while the use of **HA-5** led to the formation of 44% **11a′** and 29% of **11a**.

**Dehydrocoupling of phosphines with hydrazobenzene**. During our studies of the dehydrocoupling mechanism using **HA-2** we were surprised to discover that in the presence of hydrogenation product from **HA-2**, hydrazobenzene, with Ph₂PH (**1a**), the homodehydrocoupling product **2a** could also be observed (Fig. 6). We therefore determined whether it was possible to use hydrazobenzene as an **HA** directly. The reaction with 0.1 mmol **1a** and 0.1 mmol hydrazobenzene and 0.5 mL THF in a J. Young NMR tube was monitored using ³¹P NMR spectroscopy (Fig. 6a). It is noteworthy that the formation of **2a** took place readily at 25 °C. However, a stoichiometric amount of *t*BuOK was required for a full conversion as the gradual addition of *t*BuOK to 0.1 mmol increased the yield of **2a** to 93%.

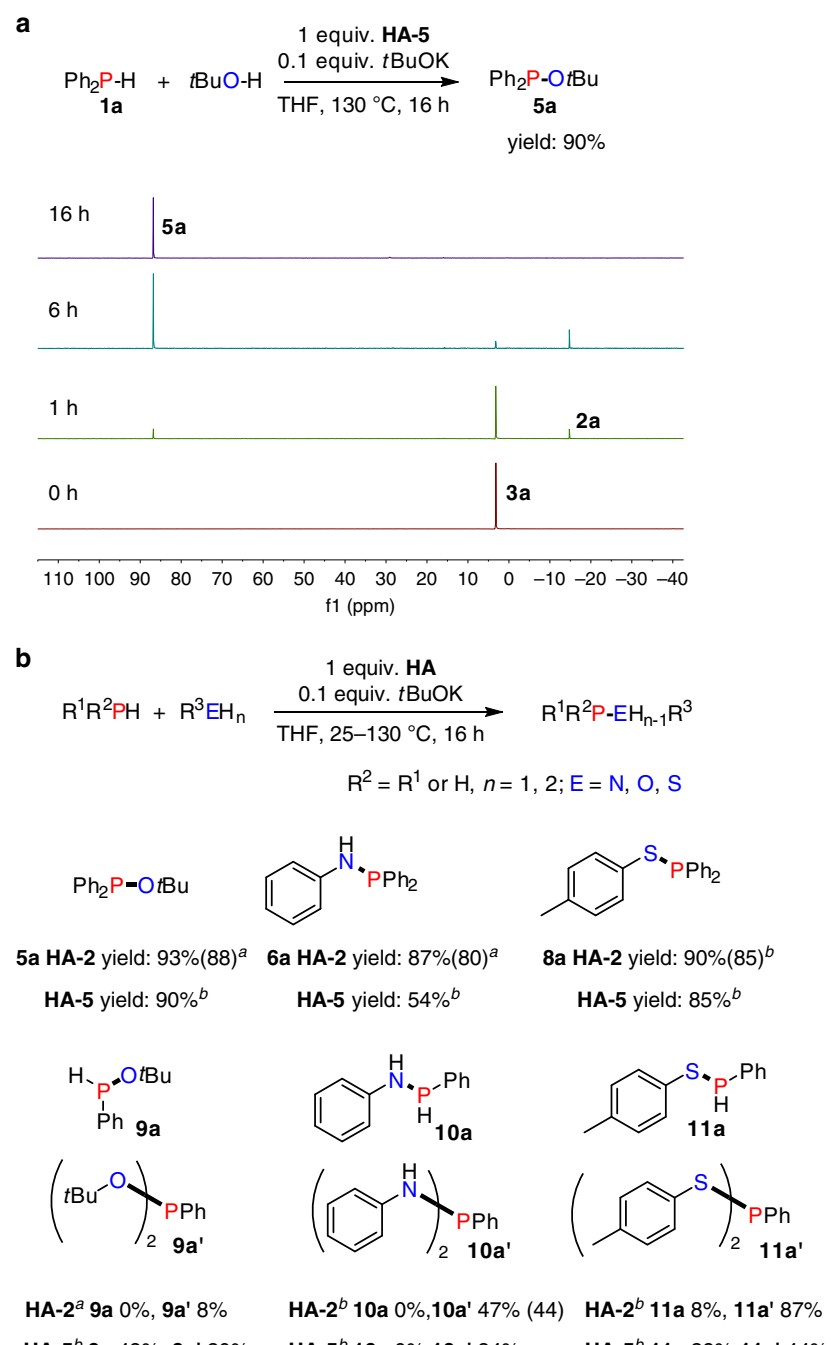

**Fig. 5** Heterodehydrocoupling of phosphines. **a** $^{31}$P{$^1$H} NMR spectra of *t*BuOK-catalysed heterodehydrocoupling of **1a** with *t*BuOH in THF. **b** Reaction generality of catalytic heterodehydrocoupling of phosphines with alcohol, amine and thiol: reactions were performed with 0.1 mmol phosphines, 0.1–0.3 mmol of R$^3$E-H$_n$; E = N, O, S, $n$ = 1, 2 and 0.01 mmol *t*BuOK, 0.1 mmol **HA-5** or **HA-2**, 0.5 mL THF in a J. Young NMR tube and yields were based on the phosphine as the limiting reagent and were determined by $^{31}$P{$^1$H} NMR spectroscopy using a capillary of PCl$_3$ as a calibration standard, the numbers in brackets were isolated yields. $^a$25 °C, $^b$130 °C

The 1:1 stoichiometric reaction of hydrazobenzene with either *t*BuOK or K[PPh$_2$] in THF at 25 °C resulted in immediate formation of a brown reaction mixture which displayed the diagnostic signal in the EPR spectrum corresponding to the K [PhNNPh], analogous to reactions where **HA-2** was used (Supplementary Fig. 29). Examples of the dehydrogenation reaction of hydrazobenzenes to azobenzenes mediated by *t*BuOK and other alkali metal compounds are known in the literature[80,81]. It therefore seems likely that in situ generation of

**HA-2** and subsequent radical species are involved in the dehydrocoupling of phosphines involving hydrazobenzene and a stoichiometric amount of *t*BuOK.

Subsequently, the generality of using hydrazobenzene in the homodehydrocoupling of phosphines was studied (Fig. 6b). We found that improved yields were obtained compared to the use of **HA-2** or **HA-5** and all reactions were completed within 5 min. Our system was tolerant to various functional groups, thus phosphines with methoxy- and methyl- and *N*,

**a**

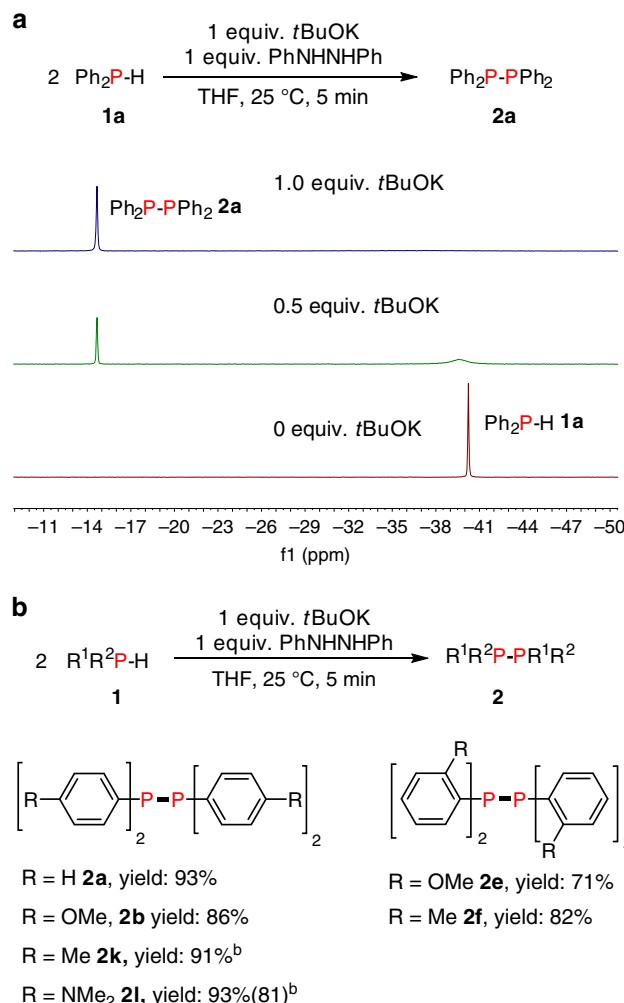

2 $Ph_2P$-H $\xrightarrow[\text{THF, 25 °C, 5 min}]{\substack{\text{1 equiv. } t\text{BuOK} \\ \text{1 equiv. PhNHNHPh}}}$ $Ph_2P$-$PPh_2$

**1a** **2a**

$Ph_2P$-$PPh_2$ **2a** 1.0 equiv. $t$BuOK

0.5 equiv. $t$BuOK

0 equiv. $t$BuOK $Ph_2P$-H **1a**

−11 −14 −17 −20 −23 −26 −29 −32 −35 −38 −41 −44 −47 −50
f1 (ppm)

**b**

2 $R^1R^2P$-H $\xrightarrow[\text{THF, 25 °C, 5 min}]{\substack{\text{1 equiv. } t\text{BuOK} \\ \text{1 equiv. PhNHNHPh}}}$ $R^1R^2P$-$PR^1R^2$

**1** **2**

R = H **2a**, yield: 93%

R = OMe, **2b** yield: 86%

R = Me **2k**, yield: 91%[b]

R = NMe₂ **2l,** yield: 93%(81)[b]

R = OMe **2e**, yield: 71%

R = Me **2f**, yield: 82%

**Fig. 6** Catalytic homodehydrocoupling of phosphines using hydrazobenzene as HA. **a** $^{31}P\{^1H\}$ NMR spectra in THF with different amounts of added $t$BuOK for the dehydrocoupling of **1a** in the presence of hydrazobenzene. **b** Catalytic results with different substituted phosphines: reactions were performed with 0.1 mmol phosphines, 0.1 mmol $t$BuOK and 0.1 mmol hydrazobenzene, 0.5 mL THF at 25 °C in a J. Young NMR tube and yields were based on the phosphine as the limiting reagent and determined by $^{31}P\{^1H\}$ NMR spectroscopy using a capillary of PCl₃ as a calibration standard, the number in brackets shows an isolated yield

N-dimethyl-substituents at *para* and *ortho*- positions all reacted well and gave the homodehydrocoupling products (**2a–b**, **2e–f**, **2k–l**) in very high yields (71–93%).

After establishing the hydrazobenzene-mediated homodehydrocoupling of phosphines, we then explored the possible extension to heterodehydrocoupling processes (Fig. 7). In general, we found that the reaction was successful, albeit with moderate product yields. For example, in the presence of 1 equiv. hydrazobenzene, diphenylphosphine reacted with aniline and produced **6a** in 52% yield. For primary phenylphosphine, doubly-dehydrocoupled products **9a′** and **11a′** were formed with $t$BuOH and *p*-thiocresol in moderate to good yield (45% and 59%, respectively).

## Discussion

We have demonstrated that cheap and commercially available $t$BuOK was capable of catalysing the homodehydrocoupling of phosphines ($R^1R^2PH$, $R^2 = R^1$ or H) in the presence of hydrogen

$R^1R^2P$-H $+$ $R^3E$-$H_n$ $\xrightarrow[\substack{t\text{BuOK 1 or 2 equiv.} \\ \text{25 °C}}]{\substack{H \\ Ph^{-N}\diagdown N^{-Ph} \\ H \quad \text{1 or 2 equiv.}}}$ $R^1R^2P$-$EH_{n-1}R^3$

$R^2 = R^1$ or H; $n = 1,2$

**6a** 52% **9a′** 45%(40) **11a′** 59%

**Fig. 7** Hydrazobenzene-mediated heterodehydrocoupling of phosphines with alcohols, thiols, and amines: reactions were performed with 0.1 mmol phosphines, 0.1–0.2 mmol $t$BuOK and 0.1–0.2 mmol hydrazobenzene in 0.5 mL THF at 25 °C for 8 h in a J. Young NMR tube and yields were based on the phosphine as the limiting reagent and were determined by $^{31}P\{^1H\}$ NMR spectroscopy using a capillary of PCl₃ as a calibration standard, the number in brackets shows an isolated yield

acceptors such as an imine or azobenzene to produce molecules containing P–P bonds. Milder reaction conditions (25 °C) and fast reaction times (<5 min) allow this process to be more practical and general for a variety of different substituted diphosphines. This is in contrast to the previously reported processes which generally use expensive transition metal catalysts or reagents requiring multi-step syntheses, harsh reaction conditions, and substantially longer reaction times. We found that imines and azobenzene play an intimate mechanistic role in facilitating the dehydrocoupling reactions rather than simply functioning as H₂-acceptors. It appears that two different reaction mechanisms operate: for imines an ionic mechanism involving a hydrophosphination adduct as a key intermediate was elucidated whereas for azobenzene, evidence for an alternative, radical coupling mechanism was apparent. In addition, saturated hydrazobenzene was also able to mediate the homodehydrocoupling reactions involving phosphines with stoichiometric amount of $t$BuOK and superior yields of the dehydrocoupled products are obtained compared to azobenzene. Significantly, a general procedure for the heterodehydrocoupling of phosphines with various main group substrates ($R^3EH_n$, E = O, S, N, $n = 1$ or 2) was also established and this provides a convenient route to compounds containing P–O, P–S, or P–N bonds. This chemistry may prove useful in the future for the preparation of phosphorus compounds with applications in biological and agrochemistry as well as in catalysis. Our current work focuses on further expansion of the substrate scope and additional detailed mechanistic studies.

## Methods

***t*BuOK-catalysed homodehydrocoupling of phosphines**. A J. Young NMR tube was charged with 0.1 mmol of the phosphines and 0.1 mmol of the corresponding hydrogen acceptors azobenzene (HA-2) or N-benzylideneaniline (HA-5), 0.5 mL of 0.02 M $t$BuOK THF solution was added to the J. Young NMR tube. The NMR tube was sealed and heated at various temperature as indicated in the corresponding Fig. 2 and Fig. 4 or Supplementary Tables 1–6 for the time shown and analysed by $^{31}P$ NMR spectroscopy. For isolated products, the solvents/volatiles were removed under vacuum and the residue was washed with hexanes, solid products were recrystallised through vapour diffusion of hexanes into THF solutions of the product.

***t*BuOK-catalysed heterodehydrocoupling of phosphines**. A J. Young NMR tube was charged with 0.1 mmol of the phosphines and 0.1 mmol of azobenzene (HA-2) or N-benzylideneaniline (HA-5), 0.5 mL of 0.02 M $t$BuOK THF solution was added to the J. Young NMR tube, then 0.1–0.3 mmol of the corresponding alcohol, thiol, or amine was added. The NMR tube was sealed and heated for the time shown in the Fig. 5 and analysed by $^{31}P$ NMR spectroscopy. For isolated products, the reactions were performed in 0.5 mmol scale of phosphine and distilled under lower pressure for liquid products or purified via diffusion crystallisation (THF/hexane).

**Homodehydrocoupling of phosphines using hydrazobenzene**. A J. Young NMR tube was charged with 0.1 mmol of the phosphines and 0.1 mmol hydrazobenzene, 0.5 mL of 0.2 M *t*BuOK THF solution was added to the J. Young NMR tube. The NMR tube was sealed and analysed directly by $^{31}$P NMR spectroscopy. The isolated products were obtained via diffusion crystallisation (THF/hexane).

## Data availability

All data to support the conclusions in this paper are available in the main text or the supplementary materials. Crystallographic data for **2j** (CCDC 1842531) and **3e**•(THF) (CCDC 1842532) are available from the Cambridge Crystallographic Data Centre. Copies of the data can be obtained free of charge from [www.ccdc.cam.ac.uk/data_request/cif].

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

## Acknowledgements

We would like to thank Dr. Saurabh S. Chitnis and Dr. Marius I. Arz for helpful discussions. We thank Paul Lawrence, Tom Leman for their help on the low temperature NMR spectroscopy experiments. We also acknowledge the EPSRC National Electron Paramagnetic Resonance Spectroscopy Research Facility and Service, and Richard Procter of Prof. Michael Ingleson's group at University of Manchester for their assistance with EPR measurements. This work was supported by European Union for Marie Skłodowska-Curie Actions for L.W. (H2020-MSCA-IF-2015_701972) and V.T.A. (H2020-MSCA-IF-2016_748371). V.T.A. also is grateful to NSERC of Canada for a postdoctoral fellowship. I.M also thanks the government of Canada for a Canada 150 Research Chair and University of Bristol for support.

## Author contributions

L.W., V.T.A., and I.M. conceived the project and designed the experiments. L.W. and V.T.A. performed the experiments and analysed the data. H.J. performed computational chemistry. V.T.A. performed X-ray crystallography. A.B. and D.C. performed EPR measurements and modelled EPR spectral data. L.W., V.T.A., H.J., and I.M. wrote the manuscript. All the authors discussed the results and commented on the manuscript. L.W. and V.T.A. share co-first authorship given their contributions indicated above.
