## [Peer Review File · Nature Communications]

Reviewers' comments:

Reviewer #1 (Remarks to the Author):

In this manuscript, Manners and coworkers describe a mild oxidative approach to P-P, P-N, P-O, and P-S bonds based on homo- or hetero- "dehydrocoupling" reactions of the P-H/X-H precursors promoted by catalytic amounts of tBuOK and stoichiometric amounts of hydrogen acceptors (HA). Logically and mechanistically, the manuscript is separated in three parts: 1) a well-defined and convincing part that outlines a reasonable catalytic cycle based on the addition/substitution reactions with the HA, 2) a speculative part where "a radical mechanism initiated by one-electron reduction of the HA by tBuOK" is proposed based on very scarce evidence, and 3) a very surprising and interesting non-catalytic coupling with stoichiometric amount of tBuOK and saturated hydrazobenzene. Overall, this is a very interesting work that reports potentially useful transformations, even though use of 1 eq. of HA is not "atom economical". I support publication of this manuscript in Nature Communications. However, a number of questions need to be answered first.

Most of my suggestions are related to the mechanism and thermodynamics of these reactions. The reaction relies on the loss of H₂. The thermodynamics of such reactions is often unfavorable (see for example, Scheme 1 in J. Amer. Chem. Soc., 2017, 139, 16210–16221). Authors need to provide the data (can be computational or experimental based on the known heats of formation) whether the 2R₂PH- \rightarrow R₂P-PR₂+H₂ reaction and similar heterocouplings is uphill or downhill. If the process is uphill, it will change the mechanistic analysis drastically. One cannot catalyze an uphill reaction and get a high yield! Most likely, the thermodynamic driving force in such case is provided by the reaction of hydrogen acceptors (HAs) and H₂.

It has to be mentioned that the mechanistic cycle in Figure 2 is not catalytic in HA.

SN₂ reaction on phosphorus is unusual. I suggest that more detailed discussion of this process is included (perhaps, in conjunction with other examples where SN₂ at a C-X bond proceeds at X instead of carbon, e.g., the halophilic reactions). In particular, the presence of three bulky groups at phosphorus in the key intermediate 3a suggests that the usual backside trajectory of SN₂ reactions would be impossible here.

It is unorthodox to suggest an unstable carbanion (6a) as a leaving group in an SN₂ reaction. Are there any precedents for that?

For the yields in Figure 2, one needs to show equivalents of HAs. Same on line 9 of page 6 – giving tBuOK in mol% and HA-5 in mmol is misleading. Give both ("0.1 mmol (1 eq)" etc).

Page 6: If I understand the description of compound 3a isomerization to the mixture of 2a, 4a and HA-5 correctly, the only way for this to happen is to include reversibility of 3a formation in the reaction scheme (modify Figure 2).

TS1 and TS2 are mentioned on page 7 but never discussed. Why mention things that are never discussed?

Evidence for the radical mechanism (pages 9-10) is weak. I understand that this is more of a hypothesis at this point but at least a reasonable mechanistic scheme needs to be suggested. By the way, it is known that the tBuOK/DMF/O₂ mixture can generate small amounts of DMF-radicals that could spark new reactivity of phosphines (J. Amer. Chem. Soc., 2017, 139, 16210–16221).

The formation of heterocoupled products under conditions where homocoupled products can be expected is interesting. It seems that there is a combination of kinetic and thermodynamic control that operates. It would be helpful for the reader if this is added to the discussion. The readers would greatly benefit from a comparison of the nucleophilicity of tBuO⁻ vs R₂P⁻ and from the comparison of the thermodynamic driving forces for the two reactions.

Minor: Personally, I would change the second line of the title "...by an Earth-abundant Alkali Metal Catalyst" to "by tBuOK". It is shorter, more informative and has no buzz words.

Reviewer #2 (Remarks to the Author):

Manners and co-workers present a base-catalyzed route to PP, PN, PS and PO bonds in the presence of a hydrogen acceptor.

The way the work is presented is misleading, not only in terms of the reactivity of the system in the manuscript but also in terms of the way the current leading examples in the literature are discussed.

The authors state wrt transition metal catalyzed phosphine dehydrocoupling that "in most cases, relatively high temperatures (110 - 140 °C) and long reaction times (3 - 4 days) were required." This is not true. As an example, Tilley's work was performed at 70 dC and took 18 h or less and (unlike this manuscript) did not require a hydrogen acceptor.

The authors state that current literature on metal-free transformations required "harsh reaction conditions and long reaction times." This is not true. As an example, Gessner's lithium carbenoid work, for the vast majority of examples, is performed at room temperature and is complete in 1 hour or less.

When discussing homo- and heterodehydrocoupling more generally the authors state "significant progress has been made in terms of the use of earth abundant metals such as Zr, Fe and Ni" but only cite two manuscripts per metal. Why were these manuscripts selected? Are these the papers the authors deem the most significant? Why focus on amine-borane dehydrocoupling? There are tens of (amine-borane) dehydrocoupling reactions using these metals alone. Why only cite one manuscript from Hill in the next section on main group catalysis. Why Hill's amine-borane publication- why not acknowledge his work on other main group dehydrocoupling reactions?

Overall the authors have not made fair representation of the literature and it does not put their own research into the proper context.

Regarding the results presented in this paper, the research is sold to the reader in terms of how mild the reaction conditions are but looking at Figure 2c there are nine products where a yield is obtained and looking more closely at the figure caption we can see that five of these reactions were performed at 130 dC for 16 h and one at 100 dC for 16 h.... these are more forcing conditions than the vast majority of published work, much of which does not require a hydrogen acceptor.

"Perhaps" side reactions had taken place when halogenated phosphines were used- this is too vague.

The intimacy of the reaction involving HA-5 and HPPH₂ and the fact that HA-5 acts as a phosphine transfer agent is interesting.

The chemistry using HA-2 is nice. But beyond this point the manuscripts reads like a collection of reactions undertaken as an afterthought and it does not contain the same level of detail as the initial studies using HA-5. This is disappointing.

The supporting information is detailed and thorough, although the validity of the two-data-point initial rate studies (one of which is the intercept with the origin!) must be questioned (figure S7).

With good knowledge of the current literature on dehydrocoupling it is difficult to see why this research is competitive, particularly when the literature is so blatantly misrepresented. This work is not suitable for publication in Nature Communications.

Reviewer #3 (Remarks to the Author):

In this contribution the authors employ hydrogen acceptors (HAs) to enable the dehydro-coupling of secondary and primary phosphines using a catalytic amount of K(O-t-Bu). While their scope studies provide a convincing argument for the operation of an ionic SN2 mechanism using an imine HA, their proposed radical mechanism employed by the azo-benzene HA allows for some useful complementarity for some bulky substrates. They then expand their reaction scope to include formation of P-O, P-N and P-S bonds via base-catalyzed heterodehydrocoupling. Interestingly, they show finally that the hydrogenated azobenzene can also serve as an HA for these reactions, albeit in the presence of stoichiometric base. While this is a solid and novel contribution, I am left with three questions: 1) Is there a dependence on the nature of the Group 1 metal that could suggest a role for Lewis-acid co-activation of the imine HA? 2) Is it more likely that t-butoxide or the diphenylphosphide anion transfers an electron to the azobenzene HA? They could discuss the relative redox potentials here. 3) Why have the authors not included a mechanistic NMR study of the use of hydrazobenzene as an HA for the dehydrocoupling reactions as they did for the other HAs? The latter is particularly important in order to assess whether this HA, which shows higher rates, could interfere with reactions employing the azobenzene HA. With appropriate answers to these questions, this work should be acceptable for publication. Several typos: p. 1, line 3 – “generally” → “often” as Ti catalysts have been employed for silane dehydrocoupling, Zr for phosphines, etc., line 4 – “cheap” → “inexpensive” (also on p. 4, line 3) as the term “cheap” tends to imply lower quality; p. 3, Fig. 1 caption, line 4 – “current work using”, para. 2, line 4 – “mediated by in situ”; p. 4, line 6 – “Homodehydrocoupling” → “Dehydrocoupling” as only the “hetero” prefix is necessary; p. 7, line 4 – “phosphide” → “diphenylphosphide”.

Responses to reviewers' comments

Our responses to the reviewers' comments below are in the order R1-R3-R2.

Reviewer 1 (R1)

R1 was very positive and concluded:

Overall, this is a very interesting work that reports potentially useful transformations, even though use of 1 eq. of HA is not "atom economical". I support publication of this manuscript in Nature Communications. However, a number of questions need to be answered first.

1.

Revision Suggested/Issue Raised:

The reaction relies on the loss of H₂. The thermodynamics of such reactions is often unfavorable (see for example, Scheme 1 in J. Am. Chem. Soc., 2017, 139, 16210–16221). Authors need to provide the data (can be computational or experimental based on the known heats of formation) whether the 2R₂PH->R₂P-PR₂+H₂ reaction and similar heterocouplings is uphill or downhill. If the process is uphill, it will change the mechanistic analysis drastically. One cannot catalyze an uphill reaction and get a high yield! Most likely, the thermodynamic driving force in such case is provided by the reaction of hydrogen acceptors (HAs) and H₂.

Our Response/Action:

We have performed thermochemical DFT calculations which indicate that the parent reaction is thermodynamically uphill, while in the presence of an HA the dehydrocoupling reactions are downhill. We have amended the main text and Supplementary Information accordingly.

In the main text (page 5) the following statement was added:

"Computationally the parent dehydrocoupling reaction of 2 equiv. 1a to 2a and H₂ in the absence of an HA was thermodynamically uphill by 2.64 kcal/mol, and likely possesses a significant kinetic barrier. The dehydrocoupling reactions with added HA-1 – HA-5 were all calculated to be thermodynamically exergonic (see Computational Section of Supplementary Information)."

2.

Revision Suggested/Issue Raised:

It has to be mentioned that the mechanistic cycle in Figure 2 is not catalytic in HA.

Our Response/Action:

We have modified the catalytic cycle in Figure 2 and have incorporated **HA-5** in the cycle, such that it is very clear that the reactions are stoichiometric and not catalytic in the **HA**.

The catalytic cycle from Figure 2b is also shown below:

3.

Revision Suggested/Issue Raised:

SN2 reaction on phosphorus is unusual. I suggest that more detailed discussion of this process is included (perhaps, in conjunction with other examples where SN2 at a C-X bond proceeds at X instead of carbon, e.g., the halophilic reactions). In particular, the presence of three bulky groups at phosphorus in the key intermediate 3a suggests that the usual backside trajectory of SN2 reactions would be impossible here.

Our Response/Action:

We agree with R1 that S_N2 at phosphorus is rare, there are several examples in the literature where experimental and theoretical studies on S_N2@P are discussed:

- Nucleophilic Substitution at Phosphorus (S_N2@P): Disappearance and Reappearance of Reaction Barriers. *J. Am. Chem. Soc.* **128**, 10738-10744 (2006). <https://pubs.acs.org/doi/abs/10.1021/ja0606529>

- Stereoselective Synthesis of P-Stereogenic Aminophosphines: Ring Opening of Bulky Oxazaphospholidines. *J. Am. Chem. Soc.* **133**, 5740-5743 (2011). <https://pubs.acs.org/doi/abs/10.1021/ja200988c>
- Nucleophilic substitution at phosphorus: stereochemistry and mechanisms. *Tetrahedron: Asymmetry* **28**, 1651-1674 (2017). <https://www.sciencedirect.com/science/article/abs/pii/S0957416617304603>

In the above examples, especially the second, S_N2 reactions at P also operate even with bulky groups on phosphorus, and the incoming nucleophile can also be iPr^- , Ph^- . The third example is a review. In the main text (page 8) we have rewritten the corresponding section as follows:

“Based on the knowledge that S_N2 reactions at phosphorus are possible⁴⁹⁻⁵¹ and similar halophilic reactions exist, we anticipated that hydrophosphination adduct **3a** is attacked by anionic **1a⁻** at the phosphorus centre through a S_N2 -type of reaction (TS1) which eventually yields **2a**. Similarly, **3a** could be attacked by the $tBuO^-$ anion (TS2) which leads to the side product $Ph_2P-OtBu$ **5a** detected by $^{31}P\{^1H\}$ NMR as a peak at 86.9 ppm (Supplementary Figure 6).”

4.

Revision Suggested/Issue Raised:

It is unorthodox to suggest an unstable carbanion (6a) as a leaving group in an S_N2 reaction. Are there any precedents for that?

Our Response/Action:

We have amended our discussion of our proposed carbanionic leaving group and included the possibility that the P-C bond cleavage may be accompanied by protonation which avoids formation of the unstable carbanion. Please note that we have renumbered some of the compounds such that the carbanion formerly known as **6a** is now referred to as **4a⁻** in the current version of the manuscript.

We also think that the benzylic effect might play a role in providing some degree of stabilization since both phenyl ring and N-atom from **4a⁻** could delocalize negative charge.

Similar carbanionic species are proposed as intermediates in Brook rearrangements (See: (a) Duff, J. M.; Brook, A. G. *Can. J. Chem.* **1977**, *55*, 2589. DOI: 10.1139/v77-358. (b) Brook, A. G.; Duff, J. M. *J. Am. Chem. Soc.* **1974**, *96*, 4692. DOI: 10.1021/ja00821a065).

Carbanionic leaving groups are also invoked in other substitution chemistry at phosphorus. (See: Capozzi, M. A. M.; Cardellicchio, C.; Naso, F. *Eur. J. Org. Chem.* **2004**, 1855. DOI: 10.1002/ejoc.200300497 and references therein).

We have amended the main text (pages 8-9) accordingly:

“Here we propose a carbanionic leaving group (**4a⁻**) which may tautomerize to the amide anion (**4a⁻**), which can also deprotonate **1a** and regenerate **1a⁻** and close the catalytic cycle. Similar transient carbanionic species are also proposed in Brook rearrangements of silylated amines in the presence of a catalytic amount of base^{52,53}, and carbanionic leaving groups are also known in reactions of phosphine oxides with organometallic reagents⁵⁴. Another possibility is following the nucleophilic attack of the phosphorus of **3a** by an anion such as **1a⁻** or *t*BuO⁻ there may be a process in which the P–C bond cleavage process is accompanied by protonation by an incoming protic substrate at the incipiently generated and partially carbanionic site.”

5.

Revision Suggested/Issue Raised:

*For the yields in Figure 2, one needs to show equivalents of HAs. Same on line 9 of page 6 – giving *t*BuOK in mol% and HA-5 in mmol is misleading. Give both (“0.1 mmol (1 eq)” etc).*

Our Response/Action:

We have revised Figure 2 and the main text accordingly such that the same units are used.

Main text (page 7) was amended as follows:

“Notably, with 0.01 mmol of *t*BuOK (10 mol%) and 0.1 mmol of **HA-5**, 82% yield of **2a** was obtained after 64 h at 25 °C”

6.

Revision Suggested/Issue Raised:

*Page 6: If I understand the description of compound **3a** isomerization to the mixture of **2a**, **4a** and **HA-5** correctly, the only way for this to happen is to include reversibility of **3a** formation in the reaction scheme (modify Figure 2).*

TS1 and TS2 are mentioned on page 7 but never discussed. Why mention things that are never discussed?

Our Response/Action:

R1 is correct, **3a** is in equilibrium with the starting imine **HA-5** and phosphine **1a**. The reversibility of **3a** formation was described in literature (Reversible P-C bond formation for saturated α -aminophosphine ligands in solution: stabilization by coordination to Cu(I). *New J. Chem.* **23**, 581-583 (1999).) and our own experiments also proved this (See supplementary materials, Supplementary Figure 1, equation 4). Following the suggestions from R1 we modified Supplementary Figure 1 (formerly known as Scheme S1) to clearly show the equilibrium, and added to the main text (on page 8).

“It is worth noting that **3a** is in equilibrium with **HA-5** and **1a** (Supplementary Figure 1, equation 4), and that the transformation of **3a** to the 1:1:1 mixture of **2a**, **4a**, and **HA-5** requires the presence of the *t*BuOK catalyst (Supplementary Figure 1, equation 5)⁴⁸.”

We mention TS1 and TS2 in the main text (page 8) as following:

“Based on the knowledge that S_N2 reactions at phosphorus are possible⁴⁹⁻⁵¹ and similar halophilic reactions exist, we anticipated that hydrophosphination adduct **3a** is attacked by anionic **1a**⁻ at the phosphorus centre through a S_N2 -type of reaction (TS1) which eventually yields **2a**. Similarly, **3a** could be attacked by the *t*BuO⁻ anion (TS2) which leads to the side product Ph₂P-O*t*Bu **5a** detected by ³¹P{¹H} NMR as a peak at 86.9 ppm (Supplementary Figure 6).”

7.

Revision Suggested/Issue Raised:

*Evidence for the radical mechanism (pages 9-10) is weak. I understand that this is more of a hypothesis at this point but at least a reasonable mechanistic scheme needs to be suggested. By the way, it is known that the *t*BuOK/DMF/O₂ mixture can generate small amounts of DMF-radicals that could spark new reactivity of phosphines (J. Amer. Chem. Soc., 2017, 139, 16210–16221).*

Our Response/Action:

All three reviewers have voiced concerns about the general lack of convincing evidence for a radical mechanism in reactions using **HA-2** as the hydrogen acceptor.

We have therefore gone ahead and performed the in-depth study originally planned for the future. This involved the use of **HA-2** in the dehydrocoupling of **1a**, and we have now also included several stoichiometric reactions where we have observed the $K[\text{PhNNPh}]$ radical anion and have demonstrated its competency in catalysis. Figure 3 of the main text is completely revised to include some of our EPR data and a proposed radical chain process. The substrate scope study with **HA-2** which was formerly in Figure 3c is now placed into a separate Figure 4 for space and clarity. Supplementary Figures 13 to 29 in the Supplementary Information are all new, and further support our proposed radical mechanism.

Please see our markedly improved section of the main text (pages 10 to 15) and Supplementary Information (Supplementary Figures 13 to 29) concerning radical chemistry of **HA-2**.

8.

Revision Suggested/Issue Raised:

The formation of heterocoupled products under conditions where homocoupled products can be expected is interesting. It seems that there is a combination of kinetic and thermodynamic control that operates. It would be helpful for the reader if this is added to the discussion. The readers would greatly benefit from a comparison of the nucleophilicity of $t\text{BuO}^-$ vs $R_2\text{P}^-$ and from the comparison of the thermodynamic driving forces for the two reactions.

Our Response / Action:

We have included computational results which indicate that the thermodynamics for P–P and P–O bond formation are similar. Kinetic control seems to be quite important and we could observe both $t\text{BuO}^-$ and $[\text{R}_2\text{P}]^-$ attack products as was described in Figure 5a and the corresponding text. The major reason why $t\text{BuO-PPh}_2$ (**5a**) was produced as the sole final product is because $\text{Ph}_2\text{P-PPh}_2$ (**2a**) was able to react with $t\text{BuOH}$ in the presence of $t\text{BuOK}$ via an alternative reaction pathway.

We have added the following in the main text (page 18):

“Computationally the heterodehydrocoupling reaction of **1a** with $t\text{BuOH}$ in the presence of **HA-5** to produce **5a** and **4a** was exergonic by 10.62 kcal/mol, similar to the computed value of $\Delta G = -10.09$ kcal/mol for the analogous homodehydrocoupling process (see Computational Section of Supplementary Information). The similarly favourable

thermodynamics calculated for the P–P and P–O coupling reactions suggest that both reactions are possible, and likely competitive. The further conversion of **2a** to **5a** in the presence of additional *t*BuOH allows for high yield of **5a**.”

9.

Revision Suggested/Issue Raised:

*Personally, I would change the second line of the title “...by an Earth-abundant Alkali Metal Catalyst” to “by *t*BuOK”. It is shorter, more informative and has no buzz words.*

Our Response / Action:

We have decided to change the title of the manuscript to:

“Homo- and Heterodehydrocoupling of Phosphines Mediated by Alkali Metal Catalysts”

Given that other alkali metal catalysts such as potassium diphenylphosphide, and radical anion K[PhNNPh] along with other alkali metal alkoxides like *t*BuONa and *t*BuOLi also facilitate reactions involving **HA-2**.

Reviewer 3 (R3)

This reviewer made very positive statements and concluded that “*While this is a solid and novel contribution, I am left with three questions...With appropriate answers to these questions, this work should be acceptable for publication..*”

1.

Revision Suggested/Issue Raised:

Is there a dependence on the nature of the Group 1 metal that could suggest a role for Lewis-acid co-activation of the imine HA.

Our Response/Action:

In Supplementary Table 4, *t*BuOLi, *t*BuONa, *t*BuOK were tested as the bases for the dehydrocoupling reactions, we did see differences, among them the efficiency followed order K>Na>Li. Lewis-acid co-activation may play a role though this was not explored in detail. Solubility was another factor noted as *t*BuOLi has very poor solubility in THF.

The following statement has been added to the main text (page 7), and a footnote is included in Supplementary Table 4:

"It is worth noting that 10 mol% of the different alkali metal *tert*-butoxides were screened as catalysts for the dehydrocoupling reaction with added **HA-5** where the activity followed the order of K>Na>Li (Supplementary Table 4)."

2.

Revision Suggested/Issue Raised:

*Is it more likely that *t*-butoxide or the diphenylphosphide anion transfers an electron to the azobenzene HA? They could discuss the relative redox potentials here.*

Our Response/Action:

Given the redox mismatch of *t*BuOK with azobenzene, and the fact that diphenylphosphide anion (**1a⁻**) is generated under the reaction conditions, and results from the stoichiometric reaction of azobenzene with **1a⁻** we think that while a small amount of radical anion K[PhNNPh] is generated in the reaction of *t*BuOK with azobenzene, diphenylphosphide is the major participant in single electron transfer chemistry.

Please see our markedly improved section of the main text (pages 10 to 15) and Supplementary Information (Supplementary Figures 13 to 29) concerning radical chemistry of **HA-2**. In particular, our detailed discussion on redox potentials (page 13).

3.

Revision Suggested/Issue Raised:

Why have the authors not included a mechanistic NMR study of the use of hydrazobenzene as an HA for the dehydrocoupling reactions as they did for the other HAs? The latter is particularly important in order to assess whether this HA, which shows higher rates, could interfere with reactions employing the azobenzene HA.

Our Response/Action:

When using hydroazobenzene in the reaction we required a stoichiometric amount of *t*BuOK. This caused the NMR studies to be very difficult as the phosphine is converted into phosphide which showed a very weak/broad peak by ³¹P NMR due to proton exchange phenomena with other species present. A comparison of reaction rates was not performed since all the reactions using azobenzene and hydroazobenzene took place at room temperature with a very fast reaction rate (oftentimes instantaneous and with complete conversion to products).

We have observed the formation of the radical anion K[PhNNPh] in reactions of hydrazobenzene with either *t*BuOK or K[PPh₂] by EPR spectroscopy, and suggest that similar radical chemistry as was seen with azobenzene is involved.

The main text (page 20) has been amended accordingly:

“The 1:1 stoichiometric reaction of hydrazobenzene with either *t*BuOK or K[PPh₂] in THF at 25 °C resulted in immediate formation of a brown reaction mixture which displayed the diagnostic signal in the EPR spectrum corresponding to the K[PhNNPh], analogous to reactions where **HA-2** was used (Supplementary Figure 29). Examples of the dehydrogenation reaction of hydrazobenzenes to azobenzenes mediated by *t*BuOK and other alkali metal compounds are known in the literature^{66,67}, and it seems likely that *in situ* generation of **HA-2** and subsequent radical species are involved in the dehydrocoupling of phosphines involving hydrazobenzene and a stoichiometric amount of *t*BuOK.”

4.

Revision Suggested/Issue Raised:

Several typos: p. 1, line 3 – “generally” → “often” as Ti catalysts have been employed for silane dehydrocoupling, Zr for phosphines, etc., line 4 – “cheap” → “inexpensive” (also on p. 4, line 3) as the term “cheap” tends to imply lower quality; p. 3, Fig. 1 caption, line 4 – “current work using”, para. 2, line 4 – “mediated by in situ”; p. 4, line 6 – “Homodehydrocoupling” → “Dehydrocoupling” as only the “hetero” prefix is necessary; p. 7, line 4 – “phosphide” → “diphenylphosphide”.

Our Response/Action:

We have revised the manuscript very carefully, and made most of the suggested changes. We have kept “generally” on page 1 line 3 as we’ve deemed this word choice is appropriate. Please see the main text for more detail, where all changes are highlighted in yellow.

Reviewer 2 (R2)

This reviewer did offer a few positive remarks:

“The intimacy of the reaction involving HA-5 and HPPH₂ and the fact that HA-5 acts as a phosphine transfer agent is interesting. The chemistry using HA-2 is nice.”

“The supporting information is detailed and thorough.”

The reviewer also made a series of critical comments which we now address.

1.

Revision Suggested/Issue Raised:

The authors state wrt transition metal catalyzed phosphine dehydrocoupling that “in most cases, relatively high temperatures (110 - 140 °C) and long reaction times (3 - 4 days) were required.” This is not true. As an example, Tilley’s work was performed at 70 dC and took 18 h or less and (unlike this manuscript) did not require a hydrogen acceptor.

Our Response/Action:

We mentioned in our manuscript that for most cases, high temperatures were required. This is true: for example, a) $[\text{Cp}^*\text{ZrH}_3]$, 120 °C, 3 days; b) $[\text{Cp}^*\text{Rh}\{\text{CH}_2=\text{CH}(\text{TMS})\}_2]$, 140 °C, 16 h; c) $(\text{N}_3\text{N})\text{ZrR}$ ($\text{N}_3\text{N} = \text{N}(\text{CH}_2\text{CH}_2\text{NSiMe}_3)_3^{3-}$), 90 – 120 °C, 7 - 36 h. The work from Tilley is exceptional, but for most of the catalytic systems high temperatures were required.

2.

Revision Suggested/Issue Raised:

The authors state that current literature on metal-free transformations required “harsh reaction conditions and long reaction times.” This is not true. As an example, Gessner’s lithium carbenoid work, for the vast majority of examples, is performed at room temperature and is complete in 1 hour or less.

Our Response/Action:

In Figure 1a of our manuscript, we have shown the reaction conditions for metal-free transformations, where in general harsh reaction conditions were required. For the work from Gessner, 0.5 equivalent of carbenoids (expensive and hard to prepare) were applied, where operation at –78 °C with warming to room temperature also be viewed as experimentally demanding reaction conditions. The major strengths we see for our methodology is the general user friendliness of the reactions given that commercially available reagents are used, and the reactions are quite general in terms of scope, and some reactions are done at 25 °C in less than 5 min.

We have also amended the statement in the main text (page 3) as follows:

“...either harsh reaction conditions, long reaction times, or non-commercially available reagents were still required.”

3.

Revision Suggested/Issue Raised:

When discussing homo- and heterodehydrocoupling more generally the authors state “significant progress has been made in terms of the use of earth abundant metals such as Zr, Fe and Ni” but only cite two manuscripts per metal. Why were these manuscripts selected? Are these the papers the authors deem the most significant? Why focus on amine-borane dehydrocoupling? There are tens of (amine-borane) dehydrocoupling reactions using these metals alone. Why only cite one manuscript from Hill in the next section on main group catalysis. Why Hill's amine-borane publication- why not acknowledge his work on other main group dehydrocoupling reactions?

Our Response/Action:

As there are a very large number of publications in especially the field of amine-borane dehydrocoupling, we tried hard to have a balanced selection of references related to different metals from different authors, and keep within the limit for number of references in the *Nature* family of journals.

4.

Revision Suggested/Issue Raised:

Regarding the results presented in this paper, the research is sold to the reader in terms of how mild the reaction conditions are but looking at Figure 2c there are nine products where a yield is obtained and looking more closely at the figure caption we can see that five of these reactions were performed at 130 dC for 16 h and one at 100 dC for 16 h.... these are more forcing conditions than the vast majority of published work, much of which does not require a hydrogen acceptor.

Our Response/Action:

For most of the published results where diphenylphosphine or phenylphosphine were used as the main substrates and very few publications studied the substrate scope. Compared to those reaction conditions using diphenylphosphine or phenylphosphines our system does require relatively mild reaction conditions. For most of the reactions from

Figure 2c which required higher reaction temperatures, we can use **HA-2** as alternative hydrogen acceptor at 25 °C where reactions were complete in less than 5 min.

5.

Revision Suggested/Issue Raised:

“Perhaps” side reactions had taken place when halogenated phosphines were used- this is too vague

Our Response/Action:

We did find that with halogenated phosphine, lower yields were obtained. We did not isolate the side products but suspect dehalogenation processes. We have changed this sentence on page 9 of the main text to:

“For chloro- and fluoro-substituted phosphines, more moderate yields were observed for **2c** and **2d**, perhaps due to the side reactions involving C–F and C–Cl bonds such as dehalogenation^{55,56”}

In addition, we cite the following papers as examples where dehalogenation of aryl halides is mediated by *t*BuOK.

Liu, W.; Hou, F. Y. *Tetrahedron* **2017**, 73, 931. DOI: 10.1016/j.tet.2017.01.002

Lin, S. B.; He, X. R.; Meng, J. P.; Gu, H. N.; Zhang, P. Z.; Wu, J. *Eur. J. Org. Chem.* **2017**, 443. DOI: 10.1002/ejoc.201601293

6.

Revision Suggested/Issue Raised:

The intimacy of the reaction involving HA-5 and HPPH2 and the fact that HA-5 acts as a phosphine transfer agent is interesting. The chemistry using HA-2 is nice. But beyond this point the manuscripts reads like a collection of reactions undertaken as an afterthought and it does not contain the same level of detail as the initial studies using HA-5. This is disappointing.

Our Response/Action:

We have added substantial EPR experiments and other mechanistic studies using **HA-2** which highlight its role in mediating radical chemistry, we believe that these further

experiments lend extensive support for the proposed radical process. We have added those results in the main text as well as the SI and these additions are highlighted in yellow.

Please see our markedly improved section of the main text (pages 10 to 15) and Supplementary Information (Supplementary Figures 13 to 29) concerning radical chemistry of **HA-2**. We found that the comparison of yield, reaction conditions, and substrate scope between **HA-2** and **HA-5** especially was quite interesting, and we viewed that the complementarity between using these different **HAs** with the same catalyst to be important conclusions.

7.

Revision Suggested/Issue Raised:

The supporting information is detailed and thorough, although the validity of the two-data-point initial rate studies (one of which is the intercept with the origin!) must be questioned (figure S6).

Our Response/Action:

For Figure S6 (now referred to as Supplementary Figure 7) we compared the initial reaction rates using differently substituted imines as **HAs**. The reaction rates were substantially different, so we chose one time point where all the products yields could be obtained. We halted the three reactions after 8 h and the reaction rate were calculated at this point. While additional data points would allow for a greater degree of accuracy for the determination of precise reaction rates, the major conclusion that was drawn from this experiment was a general qualitative comparison of the relative rates of one derivative to the next.

Reviewers' comments:

Reviewer #1 (Remarks to the Author):

My main suggestions/concerns have been addressed by the authors. The only remaining concern is the lack of citations to the closely related work. For example, the recent report on the N-C heterodehydrocoupling in the presence of tBuOK and an oxidant (J. Amer. Chem. Soc., 2017, 139, 16210–16221.) should be cited, especially since this highly relevant paper spells out how the use of the oxidant can fix the problem of unfavorable thermodynamics for the title transformation (something that authors specifically mention in the revised version).

Reviewer #2 (Remarks to the Author):

The authors have made some attempt to address the points raised by reviewers. There is much data of interest now contained in the supporting information, which would raise the question of whether this work is now more suited to publication as a full paper rather than a comm.

Reviewer #3 (Remarks to the Author):

This paper describes the use of hydrogen acceptors (HAs) to enable the K(O-t-Bu)-catalyzed homo- and hetero-dehydrocoupling of phosphines. Although certainly not as atom-economic as previously reported transition metal-catalyzed variants, their route may be especially valuable for small basic phosphines that could potentially even deactivate (or at least significantly slow down) Zr catalysts. A larger issue that the authors do not address sufficiently is the potential importance of their products. For the P-P bonded products they cite a review (ref. 30) covering addition of P-P bonds to alkenes and alkynes to make bis-phosphines without informing the reader as to its content. With regard to the P-O, P-N, and P-S products, they do not tell us what potential advantages their route may have over the universal chlorophosphine + E-H + NEt₃ procedure. While I understand that use of TM catalysts comes with some disadvantages, their route is *stoichiometric* in the HA, and they offer no clever design of the latter that would allow for its facile separation from the products. So with an inefficient reaction affording products of limited import, the main contribution of this work would have to be its novel reaction mechanisms. Using the imine HA, the key reaction step is the S_N2 attack of the diphenylphosphide anion on the tertiary phosphorus center that also bears a lone pair, as described in general in ref. 51 (refs. 49 and 50 are not relevant as they involve attack on P^(V) or P^(III) → Lewis acid). On p. 9 they note that acceleration of the rate by inclusion of an electron-withdrawing group on the imine nitrogen “provides further support for the proposed reaction mechanism.” More details would be appropriate here; i.e., does this suggest that the S_N2 reaction is the rate-determining step and that the EW group stabilizes the anionic leaving group? In Supplementary Figure 6 they obtain a time-course reaction at 50°C but do not plot the product concentration as a function of time which would have been useful in justifying the data in Supplementary Figure 7 wherein they estimate the initial reaction rates on the basis of *one observation* after 8 h at RT. Without also knowing the conversion of **1a** one cannot estimate how many half-lives the reaction has proceeded. There were also some omissions in the Experimental Section with regard to the primary phosphine reactions. For example, if the 29% yield of the 4-membered ring compound (PCyH)₄ was determined by P-31 NMR after reaction at 130°C, was this at full conversion of PCyH₂ and how were the crystals obtained? Similarly, they report an isolated yield from the phenylphosphine reaction with no details of conversion, time, etc. For HA-2 they did an excellent job on the EPR data and I think that this mechanistic work carries the paper. They could cite a relevant paper on electron transfer from phosphide anions; i.e., S. S. Wreford et al. *J. Org. Chem.* 1977, 42, 3247. After some efforts to address the above points, this work should be suitable for publication in *Nature* although the amount of work done is fairly extensive for a communication!

Reviewer 1 (R1)

R1 encouraged us to cite more references on the related work: My main suggestions/concerns have been addressed by the authors. The only remaining concern is the lack of citations to the closely related work.

Our Response/Action:

We have cited more references on the dehydrocoupling of phosphines and *t*BuOK catalyzed reactions, which are listed below are highlighted in yellow in the references of the main text.

1. Roering, A. J., MacMillan, S. N., Tanski, J. M. & Waterman, R. Zirconium-Catalyzed Heterodehydrocoupling of Primary Phosphines with Silanes and Germanes. *Inorg. Chem.* **46**, 6855-6857 (2007).
2. Rossin, A. & Peruzzini, M. Ammonia–Borane and Amine–Borane Dehydrogenation Mediated by Complex Metal Hydrides. *Chem. Rev.* **116**, 8848-8872 (2016).
3. Han, D., Anke, F., Trose, M. & Beweries, T. Recent advances in transition metal catalysed dehydropolymerisation of amine boranes and phosphine boranes. *Coord. Chem. Rev.* **380**, 260-286 (2019).
4. Weickgenannt, A. & Oestreich, M. Potassium *tert*-Butoxide-Catalyzed Dehydrogenative Si–O Coupling: Reactivity Pattern and Mechanism of an Underappreciated Alcohol Protection. *Chem. Asian. J.* **4**, 406-410 (2009).
5. Evoniuk, C. J. *et al.* Coupling N–H Deprotonation, C–H Activation, and Oxidation: Metal-Free C(sp³)–H Aminations with Unprotected Anilines. *J. Am. Chem. Soc.* **139**, 16210-16221 (2017).

Reviewer 2 (R2)

This reviewer is satisfied of our previous revisions and made comments: There is much data of interest now contained in the supporting information, which would raise the question of whether this work is now more suited to publication as a full paper rather than

a comm.

Our Response/Action:

If we understand correctly, there is only one type of research article in *Nature Communications* regardless the length of the manuscript, our manuscript is also within *Nature Communications*' guidelines regarding length.

Reviewer 3 (R3)

This reviewer offered several positive remarks:

their route maybe especially valuable for small basic phosphines that could potentially even deactivate (or at least significantly slow down) Zr catalysts.

and

For HA-2 they did an excellent job on the EPR data and I think that this mechanistic work carries the paper.

and R3 concluded that:

After some efforts to address the above points, this work should be suitable for publication in *Nature* although the amount of work done is fairly extensive for a communication!

1.

Revision Suggested/Issue Raised:

A larger issue that the authors do not address sufficiently is the potential importance of their products. For the P-P bonded products they cite a review (ref. 30) covering addition of P-P bonds to alkenes and alkynes to make bisphosphines without informing the reader as to its content. With regard to the P-O, P-N, and P-S products, they do not tell us what potential advantages their route may have over the universal chlorophosphine + E-H + NEt₃ procedure.

Our Response/Action:

We have now cited a number of papers about the applications of compounds featuring P-P bonds to highlight the potential importance of these products, these papers are listed below and highlighted in yellow in the reference list in the main text.

- 1 Geier, S. J. & Stephan, D. W. Activation of P(5)R(5) (R = Ph, Et) by a Rh-beta-diketimate complex. *Chem. Commun.*, 2779-2781, doi:10.1039/b803277g (2008).

- 2 Greenberg, S. & Stephan, D. W. Stoichiometric and catalytic activation of P–H and P–P bonds. *Chem. Soc. Rev.* **37**, 1482-1489, doi:10.1039/B612306F (2008).
- 3 Molitor, S., Mahler, C. & Gessner, V. H. Synthesis and solid-state structures of gold(I) complexes of diphosphines. *New J. Chem.* **40**, 6467-6474, doi:10.1039/c6nj00786d (2016).
- 4 Annibale, V. T., Ostapowicz, T. G., Westhues, S., Wambach, T. C. & Fryzuk, M. D. Synthesis of a sterically bulky diphosphine synthon and Ru(II) complexes of a cooperative tridentate enamide-diphosphine ligand platform. *Dalton Trans.* **45**, 16011-16025, doi:10.1039/c6dt02352e (2016).
- 5 Chitnis, S. S. *et al.* Addition of a Cyclophosphine to Nitriles: An Inorganic Click Reaction Featuring Protio, Organo, and Main-Group Catalysis. *Angew. Chem. Int. Ed.* **56**, 9536-9540, doi:10.1002/anie.201704991 (2017).
- 6 Wu, L. P., Chitnis, S. S., Jiao, H. J., Annibale, V. T. & Manners, I. Non-Metal-Catalyzed Heterodehydrocoupling of Phosphines and Hydrosilanes: Mechanistic Studies of B(C₆F₅)₃-Mediated Formation of P-Si Bonds. *J. Am. Chem. Soc.* **139**, 16780-16790, doi:10.1021/jacs.7b09175 (2017).
- 7 Feldmann, K. O. & Weigand, J. J. P-N/P-P Bond Metathesis for the Synthesis of Complex Polyphosphanes. *J. Am. Chem. Soc.* **134**, 15443-15456, doi:10.1021/ja305406x (2012).
- 8 Arisawa, M. & Yamaguchi, M. Rhodium-catalyzed addition reaction of diphosphine disulfide to aldehydes and ketones. *Tetrahedron Lett.* **50**, 3639-3640, doi:10.1016/j.tetlet.2009.03.135 (2009).
- 9 Hajdok, I., Lissner, F., Nieger, M., Strobel, S. & Gudat, D. Diphosphination of Electron Poor Alkenes. *Organometallics* **28**, 1644-1651, doi:10.1021/om801179k (2009).
- 10 Sato, Y., Kawaguchi, S.-i., Nomoto, A. & Ogawa, A. Highly Selective Phosphinylphosphination of Alkenes with Tetraphenyldiphosphine Monoxide. *Angew. Chem. Int. Ed.* **55**, 9700-9703, doi:10.1002/anie.201603860 (2016).

In regard to the advantages of our process compared to the universal chlorophosphine + E-H + NEt₃ procedure, **our method is catalytic and does not produce a stoichiometric amount of salt as byproduct**, in principle the hydrogenated products from the imine or azobenzene could potentially be recycled after the dehydrogenation process but this was not undertaken.

In the beginning of the Introduction of our manuscript, we have stated, “When compared to the traditional stoichiometric salt metathesis and reductive coupling reactions that still dominate the formation of element-element bonds in main group chemistry, catalytic methods represent a highly attractive alternative synthetic approach.”

Apart from the above points, we want to highlight that our process is, to the best of our knowledge, the second system for the catalytic dehydrogenative coupling of P-H groups with protic E-H bonds, the first being the Rh catalyst system reported by Tilley (*J. Am. Chem. Soc.* **128**, 13698-13699 (2006)).

2.

Revision Suggested/Issue Raised:

On p. 9 they note that acceleration of the rate by inclusion of an electron-withdrawing group on the imine nitrogen “provides further support for the proposed reaction mechanism.” More details would be appropriate here; i.e., does this suggest that the SN2 reaction is the rate determining step and that the EW group stabilizes the anionic leaving group?

Our Response/Action:

We have revised this section as follows:

“These kinetic observations provide further support for the proposed reaction mechanism described in Fig. 2b as an electron-withdrawing group should stabilize the anionic leaving group **4a⁻**.”

3.

Revision Suggested/Issue Raised:

In supplementary Figure 6 they obtain a time-course reaction at 50°C but do not plot the product concentration as a function of time which would have been useful in justifying the data in Supplementary Figure 7 wherein they estimate the initial reaction rates on the basis

of one observation after 8 h at RT. Without also knowing the conversion of **1a** one cannot estimate how many half-lives the reaction has proceeded.

Our Response/Action:

Our main message we wanted to convey with Supplementary Figure 6 was that during the reaction there were no other *P*-containing species involved as reaction intermediates apart from the adduct **3a**. In response to R3, we have also added a conversion vs time plot as Supplementary Figure 6b.

For Supplementary Figure 7 we compared the initial reaction rates using differently substituted imines as **HAs**. The reaction rates were substantially different, so we chose one time point where all the products yields could be obtained. We halted the three reactions after 8 h and the reaction rates were calculated at this point. While additional data points would allow for a greater degree of accuracy for the determination of precise reaction rates, the major conclusion that was drawn from this experiment was a general qualitative comparison of the relative rates of one derivative to the next. The conversion of **1a** is shown in the following figure.

4.

Revision Suggested/Issue Raised:

There were also some omissions in the Experimental Section with regard to the primary phosphine reactions. For example, if the 29% yield of the 4-membered ring compound

(PCyH)₄ was determined by P-31 NMR after reaction at 130 °C, was this at full conversion of PCyH₂ and how were the crystals obtained? Similarly, they report an isolated yield from the phenylphosphine reaction with no details of conversion, time, etc.

Our Response/Action:

Using the imine HA-5 the 29% NMR-determined yield of (PCy)₄ was obtained after 16 h at 130 °C, the CyPH₂ substrate is fully converted at this point in the reaction but there are other *P*-containing products which also form. On the other hand, when HA-2 is used (PCy)₄ was the main product obtained in 1 h at 130 °C where the NMR yield was 66%, and the isolated yield post recrystallization from THF/hexanes was 60%, we have now clarified the reaction time in Fig. 4.

When PhPH₂ was used as the substrate, using HA-2 gave clean conversion to cyclic (PPh)₅, while HA-5 gave several products after 16 h at 130 °C but nevertheless (PPh)₅ was successfully isolated in 60% from the reaction mixture through recrystallization.

Crystalline solids (PCy)₄ and (PPh)₅ were isolated by vapour diffusion of hexanes into a THF solution of the reaction mixture, we mentioned this under the 'General procedure for the *t*BuOK-catalyzed homodehydrocoupling of phosphines' (pgs. S2-S3) and the 'X-ray Crystallography' (pg. S4) sections of the Supplementary Information.

5.

Revision Suggested/Issue Raised:

For HA-2 they did an excellent job on the EPR data and I think that this mechanistic work carries the paper. They could cite a relevant paper on electron transfer from phosphide anions; i.e., S. S. Wreford et al. *J. Org. Chem.* 1977, 42, 3247.

Our Response/Action:

We thank R3 for the comment and suggestion, we have cited the above reference, we highlight this in yellow in the reference list in the main text.

REVIEWERS' COMMENTS:

Reviewer #3 (Remarks to the Author):

The authors have done a good job of addressing the reviewers' comments and their work is now suitable for publication.